# Vitreous Substitutes from Bench to the Operating Room in a Translational Approach: Review and Future Endeavors in Vitreoretinal Surgery

**DOI:** 10.3390/ijms24043342

**Published:** 2023-02-07

**Authors:** Filippo Confalonieri, Natasha Josifovska, Gerard Boix-Lemonche, Ingar Stene-Johansen, Ragnheidur Bragadottir, Xhevat Lumi, Goran Petrovski

**Affiliations:** 1Department of Ophthalmology, Oslo University Hospital, Kirkeveien 166, 0450 Oslo, Norway; 2Center for Eye Research and Innovative Diagnostics, Department of Ophthalmology, Institute for Clinical Medicine, University of Oslo, Kirkeveien 166, 0450 Oslo, Norway; 3Department of Biomedical Sciences, Humanitas University, Via Rita Levi Montalcini 4, Pieve Emanuele, 20090 Milan, Italy; 4Eye Hospital, University Medical Centre Ljubljana, Zaloška cesta 7, 1000 Ljubljana, Slovenia; 5Department of Ophthalmology, University of Split School of Medicine and University Hospital Centre, 21000 Split, Croatia

**Keywords:** vitreous substitutes, vitreoretinal surgery, retinal detachment, proliferative vitreoretinopathy, silicone oil

## Abstract

Vitreous substitutes are indispensable tools in vitreoretinal surgery. The two crucial functions of these substitutes are their ability to displace intravitreal fluid from the retinal surface and to allow the retina to adhere to the retinal pigment epithelium. Today, vitreoretinal surgeons can choose among a plethora of vitreous tamponades, and the tamponade of choice might be difficult to determine in the ever-expanding range of possibilities for a favorable outcome. The currently available vitreous substitutes have disadvantages that need to be addressed to improve the surgical outcome achievable today. Herein, the fundamental physical and chemical proprieties of all vitreous substitutes are reported, and their use and clinical applications are described alongside some surgical techniques of intra-operative manipulation. The major upcoming developments in vitreous substitutes are extensively discussed, keeping a translational perspective throughout. Conclusions on future perspectives are derived through an in-depth analysis of what is lacking today in terms of desired outcomes and biomaterials technology.

## 1. Introduction

A plethora of retinal conditions exist that require pars plana vitrectomy (PPV) and subsequent injection of a vitreous substitute into the vitreous chamber such as retinal detachment (RD), proliferative diabetic retinopathy, macular holes and ocular trauma [1,2,3,4,5]. Among these, RDs treated with PPV are the cornerstone condition in which all the properties of an artificial, temporary vitreous substitute come into play for a successful clinical and anatomical result [6,7].

Vitreous substitutes can be divided, according to their use, into those that are both applied and removed intra-operatively, and those that are left in the eye postoperatively, and either disappear on their own or require a second operation for removal [8]. Balanced salt solution (BSS), air, gases and silicone oils (SiOs) are all commonly used vitreous substitutes with different chemical and physical properties that bring along advantages and disadvantages [9]. Vitreous substitutes are generally characterized by their intraocular tamponade capability, and they are also commonly referred to as “vitreous tamponades” or “tamponading agents” [7]. More precisely, a vitreous substitute can be defined as a tamponading agent if it displays the ability to displace the water solution from a retinal break, thereby allowing for its closure after laser retinopexy or cryopexy, and maintaining the adhesion between the retina and retinal pigment epithelium (RPE) (Figure 1) [10,11].

Herein, we discuss the basic physico-chemical proprieties of all vitreous substitutes in use to date through a translational approach, in an attempt to bridge the gap between basic science and clinical ophthalmology and vitreoretinal surgery. The clinical uses and some intra-operative techniques, as well as the major upcoming developments in the field, are discussed in detail in regards to the significant forthcoming advancements. Surgeons can choose the best vitreous substitute only through in-depth knowledge of physical and chemical properties to address the pathophysiology of retinal diseases which, combined with the skills of the basic scientist, can lead to further advancements in the discipline of vitreoretinal surgery.

## 2. The Anatomy of the Vitreous Body: General Principles

The vitreous body has no or very little ability of regenerating itself after surgical removal [12]. In a normal-sized eye, the vitreous body has a volume of approximately 4 mL, making the vitreous chamber by far the most voluminous compared to the anterior chamber (0.2 mL) and the posterior chamber (0.05 mL) [13]. The vitreous body can be subdivided into three anatomical regions: the vitreous core, the vitreous base, and the vitreous cortex. The **vitreous core** forms most of the volume of the vitreous body. Within this region, the collagen fibrils course in a posterior-to-anterior direction and they blend with those of the vitreous base (where the vitreous body is most adherent to the underlying retina and ciliary epithelium) and the vitreous cortex anteriorly and posteriorly, respectively [14]. The **vitreous base** and the **vitreous cortex** both contain hyalocytes and densely packed collagen [15]. These structures are connected from anterior to posterior, the vitreous base with the *ora serrata* region and the vitreous cortex with the neurosensory retina [16].

The vitreous chamber is delimited from anterior to posterior, by the posterior face of the lens, the zonules, the *pars plicata*, *pars plana*, *ora serrata* and retina, up to the posteriorly located macula and optic nerve; the vitreous body must therefore take the form of the walls of the adjacent structures. In this tortuous course, many anatomical and virtual spaces are created that are crucial for both an accurate surgical dissection and a physiologic postoperative reconstruction of the vitreous chamber (Figure 2).

The physiological function of the vitreous body is directly correlated to its physical and chemical properties and involves supporting adjacent posterior segment structures both mechanically and metabolically, absorbing traumatic injury forces exerted onto the globe, providing an ocular refractive medium, transporting metabolites, such as oxygen, and acting as a cell barrier to inhibit cell migration from the retina to the vitreous cavity [18]. Some of these characteristics are lost with aging through occurrence of synchysis and syneresis processes [19,20]. In adult eyes, posterior precortical vitreous pockets were found by Kishi and Shimizu, who also described their function in several vitreoretinal interface conditions, although a thorough understanding of these structures remains only partially known [21,22].

The Cloquet’s or hyaloid canal is an embryonic sheath that surrounds the hyaloid vessels from the optic disc to the lens, providing nutrition and allowing for its growth [23]. In the adult eye, some remnants may persist, ranging from evident signs such as persistent fetal vasculature, Bergmeister’s papilla and Mittendorf’s dot, to minor residues, so much so that in the majority of eyes, the hyaloid artery remnants and their related components do not entirely vanish [24].

All these functions, structures, and embryologic or physiologic remnants must be taken into consideration when replacing the vitreous body with an artificial one. This complex anatomy is likely the result of a long evolution process towards the best possible visual acuity development and preservation, and must not be ignored [25]. In fact, to reproduce the physiologic vitreous body, it is believed that the ideal vitreous substitute should have the following characteristics [19,26,27]:Viscosity: 300–2000 cP;pH: 7.40–7.52;Density: 1.0053–1.008 g/cm^3^;Intraocular pressure (IOP) maintenance to less than 20 mmHg;Osmolality: 304 mOsm;Hydrophilic and insoluble in water;Easy to remove after surgery, if required;Provide support to intraocular tissues and maintains its proper position;Allow proper diffusion of ions, electrolytes, and other molecules (oxygen, lactic acid, and ascorbic acid);Easy to inject through a small syringe during vitrectomy surgery;Inert;Biocompatible;Transparent;Non-absorbable if required to be kept indefinitely or for a very long time.

The vitreous humor is relatively idle, and unlike the fluid in the anterior part of the eye (aqueous humor) which is constantly replaced, its structure is rather stable throughout the lifetime. It is a specialized type of highly hydrated extracellular matrix (ECM) with a water content of between 98% and 99.7% [14,28].

The vitreous gel, which composes most of the vitreous body, is constituted of intricate networks of collagen fibrils and glycosaminoglycans (GAGs). The vitreous is “inflated” by the osmotic contribution of the GAGs, and is given a strong structure by the collagen fibrils of types II, V, IX, and XI, with type II being the most prevalent and representing 60–75%. The amount of type IX collagen in the vitreous has been estimated to be up to 25%, while type V/XI collagen is a minor (only 10–25% of the total collagen) fibril-forming collagen that co-assembles with type II collagen to form the core of the heterotypic fibrils [14,29,30,31,32].

The GAGs and proteoglycans of the vitreous contain hyaluronic acid (HA), small amounts of the sulphated GAGs, chondroitin sulphate proteoglycans (CS), and Heparan Sulphate proteoglycans (HS).

HA is the predominant GAG in the mammalian vitreous found with the highest concentration in the posterior vitreous cortex, and while HS proteoglycans are present during development, postnatal eyes have relatively low quantities of these proteins [14].

The vitreous contains a small number of cells, mostly located in the vitreous cortex. Hyalocytes are 90% of these cells, while fibroblasts are the remaining 10%. Hyalocytes are located both anterior to the ciliary epithelium and posterior to the vitreoretinal interface. Depending on the activity and location, the hyalocytes can be round, oval, spindle-shaped, or flattened, containing a well-developed Golgi complex, moderate amounts of mitochondria, basophilic periodic acid-Schiff (PAS)-positive granules, lysosomes, and smooth and rough endoplasmic reticulum [33,34].

Hyalocytes are thought to actively contribute to the maintenance of vitreous transparency, avascularity, and production of vitreous ECM proteins, according to recent research. Furthermore, it was suggested that hyalocytes may also control intraocular inflammation to maintain the vitreous clarity [35]. According to histology, the hyalocytes resemble macrophages in their features. Lysosomes, mitochondria, ribosomes, and micropinocytotic vesicles are present in them, and they express the leukocyte-associated antigens CD45, CD11a, CD64, and S-100 as well as the major histocompatibility complex classes I and II. However, they do not express CD68, CD11b, CD14, glial fibrillary acidic protein (GFAP), or cytokeratin. Hyalocytes that resemble macrophages are also important immunity and inflammatory regulators in the vitreous cavity, where they play a role in the development of uveitis, proliferative diabetic retinopathy, and proliferative vitreoretinopathy [36,37,38].

## 3. Physical Properties of the Currently Available Vitreous Substitutes and Clinical Correlates

In the context of RD repair, the aims of tamponading agents are threefold. The first objective is to displace the liquefied vitreous from the retinal tear or hole, thus allowing for a restoration of fluid-free subretinal space aided by the pumping capability of the RPE [39]. The second objective is to allow for the development of a retinochoroidal adhesion and subsequent retinopexy through postoperative direct retinochoroidal contact [40]. The third objective is to limit the diffusion of proliferative and inflammatory cytokines into the vitreous cavity, so to prevent proliferative vitreoretinopathy (PVR) development, and at the same time supporting the retina in the long term. This effect is typically achieved with SiO [41,42].

To better understand and describe the mechanisms of action of vitreous substitutes and their intraocular behavior, the following physical properties need to be taken into account [43,44,45].


**Specific Gravity and Density**


In physics, the specific gravity is the ratio between the density of an object and a reference substance. On the other hand, an object’s density is a measure of how compact or heavy the object is, in each unit of volume. The reference substance is usually water, which has a density of 1 g/mL or 1 g/cm^3^ [46]. Another important distinction should be made between the weight and density of a tamponading agent: while weight is influenced by gravity, density is an intrinsic property. Regardless of gravity, a substance’s density remains constant. Instead, weight is a function of gravity and not a feature of matter because it depends on the gravitational field. Therefore, a material’s specific gravity is also an inherent characteristic [47].

These concepts can be applied to vitreoretinal surgery. It should be noted that the specific gravity can determine two fundamental variants in the tamponading agents’ behavior. The first is the shape of the tamponading agent bubble, in turn secondary to the buoyancy and pressure exerted on the walls of the eye. The second is the property of sinking or floating in the reference substance, which is the aqueous humor (also sometimes called “liquefied vitreous”) filling up the vitreous chamber after PPV. Substances with specific gravity higher or lower than 1.00 g/mL will either sink or float in water, respectively. According to these two features, the shape of the bubble could theoretically be the same at specular density value over and under the density of the aqueous humor, with the difference that the floating bubble will be located upwards and the sinking one downwards, tamponading opposite areas of the retina.

Aqueous humor’s specific gravity can be considered negligibly higher than water, with a specific gravity of 1.00 g/mL at exactly 4 degrees Celsius. Therefore, a vitreous substitute with a density equal to 1.00 g/mL is neutrally buoyant in water and this would not be considered a tamponading agent, strictly speaking. If compared to water, the specific gravity of the vitreous humor is also slightly and negligibly higher [26,48].

As far as the currently available tamponading agents are involved, the specific gravity of SiO is 0.97 g/mL, making it slightly buoyant. On the contrary, the specific gravities of perfluorohexyloctane (F_6_H_8_) (1.35 g/mL) and perfluorodecaline (F_6_H_10_) (1.93 g/mL) are significantly higher than the specific gravity of water [49]. Further details on the tamponading agents currently in use in vitreoretinal surgery will be discussed below.

b.
**Buoyancy**


In physics, the weight of an object is the force acting on the object due to gravity [50,51]. Buoyancy (also known as the buoyant force) is the force exerted on an object that is wholly or partly immersed in a fluid. Archimedes’ principle states that the upward buoyant force that is exerted on a body immersed in a fluid, whether fully or partially, is equal to the weight of the fluid that the body displaces [52].

Weight force and buoyancy force, hence, act on an immersed object in the opposite directions. In the same way, an intraocular bubble of a tamponade agent is acted upon by two opposing forces: buoyancy (upward force) and weight (downward force). The result is the force with which the bubble presses against the retina. Therefore, the more a tamponading agent floats or sinks, the more pressure it exerts on the retinal area on which it acts. This has multiple useful functions that the vitreoretinal surgeon can exploit to displace subretinal fluid, to press the retina onto the RPE and choroid allowing for rapid postoperative contact and effective chorioretinal adhesion formation, to close the contact between the aqueous humor filling up the vitreous chamber and the subretinal space (allowing time to the RPE to further get rid of the subretinal fluid), and to displace proliferative and inflammatory cytokines from entering in contact with the area of the retina that is in contact with the vitreous substitute. This last characteristic, combined with the fact that the majority of vitreous substitutes float, produces the phenomenon of inferior PVR development.

It is therefore apparent that SiO has a relatively little buoyant force when immersed in aqueous humor, as their specific gravity is similar. On the contrary, the buoyant force is greatest with air, as its specific gravity is the lowest among the vitreous substitutes used as of today, compared to aqueous humor (Figure 3).

c.
**Interfacial Tension**


In physics, surface tension is the property of a liquid in contact with a gas phase (usually air). Interfacial tension, on the other hand, is the property between any two substances. It could be liquid-gas, liquid-liquid, liquid-solid or solid-air [53].

Interfacial tension is defined as the amount of energy required to increase the interfacial area between two immiscible adjacent phases. The energy needed to increase the area of a substance must, in other words, overcome the internal adhesion energy generated by the Van der Waals forces. In fact, these forces are weak electrostatic ones that attract oppositely charged molecules that constitute a substance, and vitreous substitutes are no exception. Therefore, two substances in contact inside the vitreous chamber display internal molecular bonds that are generally stronger than the external molecular bonds created with the substance they are in contact with and the retinal tissue itself. Consequently, a substance with a high interfacial tension will have a greater tendency to stay as one single large bubble without dispersion into small bubbles, which is a fundamental characteristic that the vitreoretinal surgeon exploits daily both intra-operatively and postoperatively. More in detail, a tamponading agent with high interfacial tension, such as gas or perfluorocarbon liquids (PFCL), will not readily sneak into the subretinal space through retinal tears because to do so, it would need to increase its surface area. This process requires the application of a high enough energy to supersede the Van der Waals forces, and therefore it can happen only when, for example, the gravity force applies energy on a PFCL bubble sitting over a stiffened, teared retina, possibly leading to the feared complication of retained subfoveal PFCL [54,55,56].

In general, in vitreoretinal surgery, the gas phases have an interfacial tension of about 80 mN/m, the highest compared to the aqueous humor [57]. On the contrary, PFCLs and SiOs have a lower interfacial tension, about 40–45 mN/m and 35 mN/m, respectively [44].

d.
**Viscosity**


In fluid dynamics, viscosity is defined as a measure of a fluid’s resistance to flow. It is generated by both the internal friction of a moving fluid and between a flowing fluid and its container [58]. In general, a fluid with a high viscosity resists motion because its molecular makeup gives it a lot of internal friction, whereas a fluid with low viscosity flows easily because its molecules have little shearing stress when it is in motion.

In vitreoretinal surgery, a tamponading agent needs to be manipulated easily through injection and extraction without utilizing dangerous pressure, so a low viscosity comes in handy. On the other hand, the tendency of a substance to emulsify and disperse into droplets over time is also dependent on its viscosity and the less viscous a substance, the lower the energy required to disperse a large bubble of the substance into small droplets.

Among the currently available tamponading agents, SiOs have a high viscosity (1.000–5.000 centistokes (cSt; 1 cSt = 106 m^2^/s)) and, once dispersed, the small droplets tend to coalesce back to a single bubble with time. However, once coated by surfactants, the small droplets may remain dispersed and sneak through retinal tears into the subretinal space or the zonules in the anterior segment [9,19,59,60]. This phenomenon can lead to other complications such as inflammation, corneal endothelial damage, high IOP, glaucoma, optic neuropathy and even retinal toxicity [61,62].

## 4. Chemical Properties of the Currently Available Tamponades and Clinical Correlates

**Balanced salt solution (BSS).** By definition, BSS is a solution characterized by an ionic composition, pH, and osmolality similar to the aqueous humor [63]. Sodium, potassium, calcium, magnesium, and chloride are the most typical elements found in BSS [64]. To preserve the integrity of tissues and cells, BSS is mandatory during PPV, as it provides a healthy pH and osmotic pressure, while giving water and inorganic ions to the cells, and it can be left into the vitreous cavity at the end of the procedure [65]. However, since BSS is mainly made up of water and salts as aqueous humor, it may not even be considered a tamponading agent but only a vitreous substitute, without other properties than a paraphysiologic eye cavity filling effect.

**Gas.** Four different intraocular gases are commonly used and commercially available in vitreoretinal surgery: air [66], sulfur hexafluoride (SF_6_) [67], perfluoroethane (C_2_F_6_) [68] and perfluoropropane (C_3_F_8_) [69,70]. They are all non-toxic, inert gases that are insoluble in the aqueous humor and, apart from air, they are all inorganic and composed of fluoride and sulfur atoms bound together [71]. Air does not expand when injected into the eye. On the contrary, when pure SF_6_, C_2_F_6_, and C_3_F_8_ gases in the vitreous cavity enter in contact with a higher pN_2_, pO_2_, and pCO_2_ in the tissue fluids, they can expand, doubling, tripling, and quadrupling their volume, respectively (Table 1). In particular, when one of these expansile gas enters the vitreous cavity, it goes through three different phases: expansion, equilibrium, and dissolution (Figure 4) [72]. It is clear that this effect by molecular oxygen can become clinically significant only in cases associated with absorption of other molecules, and when the partial pressure of the molecular oxygen is considerably higher than the physiologic [73,74]. In the immediate postoperative period, the gas bubble absorbs nitrogen, oxygen, and carbon dioxide from the surrounding tissue fluids, which causes the first expansion. The first 6 to 8 h following gas injection are when the rate of expansion is fastest. The partial pressures in the two compartments equalize during the equilibrium phase as the diffusion of gas into the surrounding fluid balances the diffusion of nitrogen into the bubble. Finally, the gases’ volume is greatly reduced through absorption into the bloodstream until complete disappearance [75]. The time of this final process corresponds to the permanence time of gas in the vitreous chamber. Furthermore, SF_6_, C_2_F_6_, and C_3_F_8_ expand significantly in cases of low atmospheric pressure, potentially leading to high IOP and central retinal artery occlusion [76]. The same volumetric expansion can also be caused by higher-than-normal blood partial pressure of nitrous oxide, oxygen and carbon dioxide, and carbon monoxide, as in cases of nitrous oxide anesthesia, oxygen therapy, broncho pneumopathy and carbon monoxide intoxication [73,77]. To avoid the risk of gas expansion, a non-expansile concentration of gas is usually injected into the eye at the end of the surgery [75,78].

**Silicone oils (SiO) and heavy silicone oil (HSiO).** These oils are a family of hydrophobic polymeric and monomeric chemical compounds made of silicon-oxygen linkages. They are also known as organosiloxane and HSiOs differ from SiO for the combination of semifluorinated alkanes (SFAs) and the radical side groups, resulting in a tamponading effect on the inferior retina, which is what makes it ‘’heavier’’ than the aqueous humor [79]. By definition, oils are hydrophobic and have the property of repelling aqueous humor, which is important to isolate the subretinal space from the aqueous humor in the vitreous cavity once a RD is treated with PPV [80]. Hereby, both SiO’s and HSiO’s chemical properties are described.

**Silicone oils (SiOs).** Chemically, the SiOs are made up of radical side groups branching off from a central, linear chain of siloxane repeating units (-Si-O). In vitreoretinal surgery, these radical side groups are potentially hydrocarbon radicals such as methyl, phenyl, vinyl, and trifluoropropyl groups. The combinations that theoretically one can make are manyfold because one -Si-O unit can join two radical side groups that can be the same or different. For example, one can have a dimethyl-siloxane unit or, on the contrary, a vinyl-methyl-siloxane unit, where these units are repeating themselves to generate a long chain [81,82].

In vitreoretinal surgery, when it is referred to SiO, it typically means a specific high molecular weight polymer type made up of polydimethylsiloxane (PDMS). PDMS is a high molecular weight polymer where the side chains are always methyl groups. Chemically, PDMS is composed of repeating units of CH_3_[Si(CH_3_)2O] n Si(CH_3_)_3_ [83].

Theoretically, the major differences that the vitreoretinal surgeon encounters when choosing a SiO as a tamponading agent are determined by the molecular weight (MW), the linear chain’s length, the radical side group’s chemical makeup, the polymer chains’ radical end termination, and the chain’s size distribution [81,84]. Practically speaking though, PDMS is what is commercially available in the clinic and the only choice that a surgeon can make is the MW [85,86].

The MW, which is determined by the length of the polymers, determines in turn the viscosity of the SiO, which is measured in centistokes (cSt). Increasing a SiO’s or PDMS’s polymer chain length means increasing the MW, which in turn increases the viscosity and therefore it corresponds to a higher cSt value [87]. The viscosities of the most commonly used PDMSs in vitreoretinal surgery today are 1.000 cSt (MW of 37 kDa) and 5.000 cSt (MW of 65 kDa) [80,88]. Although 5.000 cSt SiO has a reduced tendency to emulsify in the lab, in vivo the 1.000 and 5.000 cSt SiOs emulsify at a similar rate, with only an apparent slight clinical advantage of 5.000 cSt SiO over the 1.000 SiO [88,89,90].

PDMS’s quality and purification is crucial for obtaining a stable and non-toxic SiO for use in vitreoretinal surgery, and it is today known that SiOs can have a low long-term toxic effect on the eye tissues, and even to the tissues surrounding the eye globe. Finding a solution to that is still an ongoing topic of research [62,91,92,93].

Furthermore, another challenge to the use of PDMS is that it is buoyant, having a specific gravity of 0.97 g/mL, whereas the vitreoretinal surgeon often has to treat inferiorly located detachments, PVR, and retinal tears [94]. To overcome this problem, PDMS and SFAs have been combined to take advantage of the high viscosity of SiO and the high specific gravity of SFAs to generate a HSiO as discussed below.

**Heavy silicone oils (HSiO).** Chemically combining SiO with SFAs results in the formation of HSiO, an umbrella term describing homogenous solutions that have a specific gravity of about 1.03–1.10 g/mL according to the type (presented below), and that therefore tend to sink in the aqueous humor [95]. HSiO has been created to treat large inferior breaks with RDs, as well as RDs complicated with inferior PVR, reRDs with inferior PVR, RDs with PVR treated with inferior retinotomies and patients unable to maintain a head-down posture [96].

Today the market offers three prefabricated, chemically different HSiOs: Densiron 68, Oxane HD, and heavier–than–water silicone oil (HWS 46-3000) [97,98].

**Densiron 68.** Perfluorohexyloctane, 30.5% SFA F_6_H_8_ and 69.5% SiO 5000 cSt are combined to create this compound (Fluoron Co, Ulm, Germany). By adding SiO, F_6_H_8_’s viscosity rises from 2.5 to over 1400 mPa·s, decreasing its tendency to disperse, which is thought to be the root of the issues associated with long-term usage of F_6_H_8_. It has a refractive index of 1.387 and a specific gravity of 1.06 g/mL [99,100].**Oxane HD.** A blend of 88.1% Oxane 5700, a 5000 mPa SiO, and 11.9% RMN_3_, a partly fluorinated olefin, makes up this compound (Bausch & Lomb, Toulouse, France). Its specific gravity is only about 1.02–1.04 g/mL. It is the least heavy HSiO of the three compounds and has the highest viscosity (3800 mPa·s) [101,102,103].**HWS 46-3000.** The most recent HSiO to reach the market is known as HWS 46-3000, and it is composed of 55% perfluorobutylhexane (F_4_H_6_), a semifluorinated alkane with viscosity of 1.28 mPa, and 45% ultrapurified SiO 100,000 (viscosity 97,100 mPa·s and specific gravity 0.977 g/cm^3^) [97]. The resultant solution is homogenous and stable in the presence of aqueous humor, air, or PFCLs, with a specific gravity of 1.105 g/mL and a viscosity of 3109 mPa·s. Of the three HSiOs, it is the heaviest and most viscous [97,98]. In a pilot trial, HWS 46-3000 was used as a long-term tamponade (1–3 months), and although it is a compound with increased viscosity, which could make this compound difficult to handle, a good success and low complication rates could be observed [98].

In the treatment of RDs secondary to inferior tears, Densiron 68, Oxane, and HWS46-3000 have demonstrated good outcomes [104]. In contrast, a prospective, multicenter, randomized controlled trial (HSiO Study) comparing the effects of heavy tamponade (Densiron 68) and conventional SiO in eyes with inferior and posterior PVR grade C or above concluded that there were no appreciable advantages to using heavy tamponade instead of conventional SiOs in these cases [105]. Likewise, some of the following studies have shown no clinically apparent advantages in utilizing HSiO over standard SiO; letting the question on the superiority of HSiO in the above-mentioned clinical circumstances still be open, above all it is important for one to consider the higher inflammation side effects brought about by SFA [104,106,107,108,109,110].

## 5. Classification of the Vitreous Substitutes: General Principles, Past and Future Attempts and Clinical Correlates

Now that both the physical and chemical properties of all the currently available vitreous substitutes have been presented, it is possible to illustrate and critically understand the classification of all the tamponading agents that have been used, proposed, and are under development. The main objective of this section is to classify the already introduced vitreous substitutes in an up-to-date clinical context and to present the intra-operative usage of PFCL and polymers that set themselves apart for its intra-operative usage (PFCL) or for its unavailability on the market (polymers).

For this purpose, vitreous substitutes can be classified into three types according to their physical status:Gases, such as air and expansile gases;Liquids, such as BSS, PFCL, SFAs and SiO;Polymers (hydrogels, smart hydrogels, and thermosetting hydrogels) [59,111].


**Gases**

**Air**
The air in the vitreous cavity is inert and colorless. Ohm utilized it for the first time to treat RD in 1911 [59,112]. Air has the advantage of being always available at no cost, but it needs to be purified via a filter before being injected into the vitreous cavity to avoid contamination [113]. Since air remains in the vitreous chamber only for a few days before being replaced by aqueous humor, it has the clinical advantage of a rapid visual rehabilitation and for this purpose has been largely utilized in the vitreoretinal surgery for RDs and macular holes [114,115,116,117,118]. However, this could be considered a disadvantage as a vitreous replacement due to its short time effect, especially in case of PVR or complex RDs. In these cases, a clear superiority of longer lasting tamponade effect has been demonstrated [119,120,121,122]. Another negative aspect of air is its low refractive index, which results in total light reflection and hence poor optical performance [123]. Air is generally used as a tamponade for RDs with upper retinal breaks [66,124,125], but it has also been recently shown to have a role in inferior retinal breaks [118].
**Expansile Gases**
Intraocular gas tamponades have been an integral aspect of vitreoretinal surgery since the early 1970s, when E.W. Norton reported the use of SF_6_ as a vitreous substitute [126]. Today, SF_6_, C_2_F_6_, and C_3_F_8_ are being used to treat a variety of vitreoretinal disorders. These gases are heavier than air, colorless, odorless, harmless and have different lasting times in the eye (Table 1). According to Kontos et al. the mean duration for 30% SF_6_ in the fluid-filled vitreous chamber is equal to 18.0 days with a standard deviation of ±2.6 days. The mean duration for 20% C_2_F_6_ is 34.5 days with a standard deviation of ±3.3 days. Finally, the mean duration for 15% C_3_F_8_ is 67.7 days with a standard deviation of ±5.5 days [75]. This is clinically important because the longer the tamponading agent persists into the eye, the longer the retina is supported, the displacement of the proinflammatory aqueous humor is carried out and the passage of fluid via the retinal breaks is blocked [1]. Within approximately a couple of days, pure SF_6_ increases to twice its initial volume, whereas C_2_F_6_ grows to roughly three times and C_3_F_8_ to about four times their respective initial volumes [127]. The knowledge of this effect is also important in surgical practice when the surgeon is confronted with subretinal fluid or choroidal effusion. The ensuing underfilling of the vitreous chamber at the end of the operation can be overcome by a slightly higher gas concentration titrated on the specific gas’s expandability [74,128,129,130]. Furthermore, it has been shown that quick variations in ambient air pressure cause significant changes in intraocular pressure and therefore patients are recommended to postpone flying travel and stay away from high altitudes for approximately 2 weeks, 4 weeks, and 6 weeks after receiving SF_6_, C_2_F_6_, and C_3_F_8_, respectively [76,131]. Extreme caution is recommended in these cases of expansile gas tamponade as the risk of post-operative ocular hypertension could lead to dreadful consequences such as central retinal artery occlusion, among others [132,133,134]. Finally, thanks to the buoyancy that characterizes gases, the surgeon can keep on treating the patient for an optimal subretinal fluid expression, as happens in the steamroller maneuver [135], for displacing the subretinal fluid from the macula (face-down positioning) [136] and for allowing an optimal retinochoroidal contact to develop into a firm adhesion. In fact, chorioretinal adhesions caused by cryopexy or laser retinopexy have been described to take between 2 and 4 weeks to reach the maximum strength of adhesion, and in order to occur, a retinochoroidal contact needs to be established and maintained [137,138,139].

**Liquids**

**Perfluorocarbon Liquid (PFCL)**
Various types of PFCLs have been used in vitreoretinal surgery, such as perfluorooctane (PFO), perfluoroperhydrophenanthrene (Vitreon), perfluorodecalin (PFD), and perfluorotributylamide (PFTB) [9,140]. The PFCLs are a class of fluorochemicals in which all the hydrogen atoms have been replaced by fluorine, and have been artificially produced to exhibit high specific gravities between 1.76 and 2.03, [140] with low surface tension and low viscosity as defining characteristics [141]. Such a high specific gravity is useful in flattening the retina up from the posterior pole, at the same time expressing the subretinal fluid out of the macula through the retinal breaks and at the same time anteriorizing the peripheral vitreous for an easier removal [142,143]. One of the main disadvantages of PFCLs is that they are approved only for intra-operative use only, since many animal studies have indicated that leaving PFCL in the vitreous cavity as a post-operative tamponade agent causes retinal toxicity [144,145,146]. It is not completely clear where PFCL’s toxicity stems from, but it has been shown that it could be a combination of induced inflammation, impurities and chemical toxicity and mechanical trauma, not only in the long term but also acutely [147,148,149,150]. Some of these impurities have been identified as molecules with nitrogen bonds, as well as compounds containing hydrogen and fluoride [151,152]. Nevertheless, there are studies reporting on the use of PFCLs as short-, medium-, and long-term vitreous substitute to exploit their high specific gravity in a way similar to HSiO [153,154,155,156,157].
**Semifluorinated Alkanes-(SFAs)**
SFAs were introduced in the early 2000s as a novel family of chemicals with excellent characteristics for application in vitreoretinal surgery [99]. They have an index of refraction of 1.3, which is similar to the aqueous humor (1.336) [158,159], they are soluble in PFCL, hydrocarbons, SiOs and contain perfluorocarbon and hydrocarbon side chains. SFAs have a specific gravity of 1.35 g/mL, are inert, and are heavier than the aqueous humor, which is the main characteristic for which they were combined with SiO [99,160]. SFAs have been used alone in the past as a temporary vitreous substitute for maximum of two to three months length of use due to their instability and inflammatory side effect [47,161,162,163]. SFAs today are used only in combination with SiO to produce HSiO and obtain a lower retinal tamponading effect [100,164].
**Silicone Oil (SiO)**
SiOs have been used as an intraocular tamponade due to its physical qualities of transparency, chemical inertness, high surface tension, and higher interfacial tension than aqueous humor since Cibis et al. introduced it in 1962 [165]. These characteristics made SiO the only material approved by the Food and Drug Administration (FDA, USA) for long-term vitreous replacement [84,86]. Moreover, SiO has a higher refractive index (1.405) as compared to the vitreous (1.336), in contrast to air and gases (about 1.0003) and the understanding of this concept is of clinical importance in the visual rehabilitation of the patient (Figure 5) [166,167].Similarly to what happens with a gas tamponade, SiO is capable of blocking intravitreal fluid from migrating into the subretinal space so that, if retinopexy has been correctly applied, chorioretinal adhesion and scarring can occur [84,86]. Once the retinochoroidal adhesion has developed and there is no longer any retinal traction, SiO can be removed [168].In contrast to gases, SiOs have a lower buoyancy, thus, the SiO bubble’s resting position cannot produce a high pressure against the retinal tear, and the sealing occurs almost exclusively as a result of the strong interfacial tension of the bubble over the break [86].The main clinical difference that sets SiO apart from gases is its ability to persist for a long time into the vitreous chamber, until surgical removal is performed [84]. This feature is crucial when contrasting the tendency of the retina to contract, shorten, and detach, a phenomenon produced by PVR or proliferative diabetic retinopathy (PDR) with residual tractions after surgery [84,169,170]. Moreover, the persistence of SiO in the vitreous chamber allows the eye to contrast the tendency to hypotony and eventually phthisis [171], thus allowing a retinal support when the retinal breaks but could not be adequately treated [172] or when the patient cannot posture properly after the surgery [84].Even though SiO has proved to be safe and effective in the above-mentioned circumstances, it is today demonstrated to have a low grade toxicity and side effects, especially in the long run [173,174]. The main mechanism causing SiO complications has been shown to be emulsification, a process that produces emulsified SiO droplets, which detach from the main SiO bubble and generate ocular inflammation, keratopathy, late-onset glaucoma, retinal toxicity, and optic neuropathy as well as loss of the tamponading capability.
**Heavy Silicone Oil (HSiO)**
After the first description of SFAs by Meinert and Roy in 2000 [99], the HSiO was created through a combination of unstable, proinflammatory SFAs with high viscosity HSiO to overcome the resting position of the buoyant or ‘’light’’ SiO. Clinically, the HSiOs are mainly utilized as an endotamponade agent for complex RDs, particularly those with inferior breaks and inferior PVR [175]. HSiO has the advantage of tamponading the retina inferiorly without the need of posturing the patient, displacing the ‘’PVR soup’’ from the bottom of the eye and therefore preventing inferior redetachments [102,104,176]. Nevertheless, HSiOs, containing the instable SFA molecules, have the disadvantages of producing inflammation and emulsion [104,177].

**Polymers**


During the last decade, new tamponade solutions have been developed in view of the existing limitations of the clinically used tamponades (hydrophobic materials such as SiO or HSiO). The development of biocompatible, biodegradable, and injectable hydrogels-based vitreous substitutes (natural, synthetic, and smart) that will also function as medium- and long-term tamponade agents has been the focus of recent studies and many such prototypes have been developed (Table 2a,b; [168,178]) [168,179]. Hydrogels-based biomaterials possess good physico-chemical properties such as high water content, optical transparency (visible range), appropriate refractive indices and density, adjustable rheological and porous properties, injectability, biocompatibility and the ability to tamponade retina on all sides through viscosity and swelling pressure [180,181]. The low toxicity of hydrogels-based biomaterials can avoid the extraction of the employed current tamponades (e.g., SiOs, HSiOs) after the chorioretinal adhesion has occurred. Many natural and synthetic polymers showed promising results as hydrophilic tamponades such as HA [182,183,184,185,186,187,188], alginate [183,189], collagen [187,190], gellan [191,192], chitosan [188,189,193], polyvinyl alcohol methacrylate [194], polyvinyl alcohol [195,196,197,198,199], poly (ethylene glycol) [200], acrylic acid [201], acrylamide [185,202], poly N-acryloyl [203] and glycinamide-polycarboxybetaine acrylamide [203]. However, the clinical application of many of the previously described materials has been hampered by their lack of transparency, deviating refractive indices, degradation, or poor biocompatibility, and even the toxicity of the crosslinking agents. Another crucial feature is the requirement for the polymer to be crosslinked. Non-crosslinked polymers have lower viscoelasticity, porosity and short residence periods, degradation, and poorer tamponade characteristics. However, due to the toxicity of the crosslinking agents that remain in the eye and the difficulty of exact control over the injection duration, the use of chemical crosslinking hydrogels is not suggested [180]. A promising alternative as a tamponade could be the use of thermosensitive hydrogels, which create a gel at higher temperatures, and recover their liquid properties at a lower temperature [204]. The absence of toxic crosslinkers allows the thermosensitive gels to be injectable in situ hydrogels with intrinsic biocompatibility [205,206]. The major challenge using thermosensitive gels as vitreous substitute, however, is how to generate a tamponading force that can enable the retina to re-adhere and guarantee the transparency within the vitreous space [207]. In addition, hydrogel-based vitreous substitutes appear to be promising carriers for drug delivery. These tamponades showed successful delivery of drugs such as Dexamethasone (HA hydrogels) [208], 5-fluorouracil (polyvinyl alcohol/chitosan hydrogels with drug loaded in poly(lactic-co-glycolic acid microspheres) [209], ascorbic acid (polyethylene glycol monomethacrylate (PEGMA)/polyethylene glycol diacrylate (PEDGA) hydrogels) [210], ascorbic acid plus glutathione (PEGMA/PEDGA hydrogels) [211], Bevacizumab (Silk/HA hydrogels) [212,213], and Bevacizumab plus aflibercept (polyethylene glycol/polypropylene glycol/polycaprolactone hydrogels) [207,214]. Despite the fact that the gentle preparation conditions and high-water content of hydrogel-based tamponades are favorable in preserving the activity of biopharmaceuticals such as peptides, proteins, or nucleic acid [215,216,217], they have not yet been employed for peptide delivery. Probably, the next step towards improving the hydrogels-based tamponades could be the use of supramolecular nanofiber hydrogels due to their ability to create hierarchically organized structures with the diversity of amino acid function such as peptide-based materials [218]. Many native peptides are able to self-assemble into filamentous networks [219,220,221]. Some authors found that a specific number of ionic self-complementary peptides that contain hydrophilic and hydrophobic side chains on the different sides of self-assembled β-sheet structures can be put together under physiological conditions into well-defined transparent nanofiber hydrogel scaffolds [222,223].

Hydrogels-based tamponades have the potential to be an excellent solution for clinical application in patients due to their proper viscosity, porosity, superior mechanical strength, and the possibility for drug encapsulation [168].

## 6. Limitations of the Current Vitreous Substitutes: Mimicking the Vitreous Body

a.
**The Physiology of Human Vitreous: What should we mimic?**


The eye’s vitreous, which is the biggest eye component in terms of volume, is crucial to the development of the eye and its structures. The biosynthesis of HA may have a significant role in the vitreous formation, which is necessary for the development of the eye and the RPE, but not the neuroretina [238]. Through the Donnan swelling process, HA may contribute to internal ocular tension preservation [239]. The eye is constantly subjected to micromovements, therefore the vitreous experiences a lot of friction, vibration, and low-frequency mechanical stress. Because the vitreous has a high concentration of HA, it behaves more like a viscoelastic body than a viscous solution, allowing for shock absorption [240]. With characteristics very close to the aqueous humor, the vitreous humor is an extremely transparent ocular material that transmits around 90% of visible and near-infrared light range [241]. Little light scattering happens in the vitreous, mainly because collagen fibers are relatively far apart due to the large HA molecules connected to them [241,242]. The lens capsule is supported by the vitreous, which also has the potential to help in accommodation [238,243]. Different biological compounds (mainly macromolecules) and cells are blocked by the vitreous [244]. The vitreous, a crucial component of the blood-ocular barrier, inhibits neovascularization, inflammation, and proliferation in its normal form [238,245,246]. The vitreous may be helpful in preventing bacterial infection and the associated inflammation, but some viral infections may be made easier by its presence. The addition of vitreous to cell culture dramatically boosts viral transduction, suggesting that the vitreous might be a good substrate for propagating the adenoviral infection process [247]. Since the late 1960s, it has been known that the vitreous can function as a metabolic reservoir for adjacent tissues [248]. Furthermore, the transfer of chemicals through the vitreous may influence the metabolism of surrounding tissues [16]. Since the vitreous reduces the oxygen exposure to the lens, its gelatinous condition may shield the lens from oxidative damage and prevent or minimize cataract development by consuming oxygen via an ascorbate-dependent process [59,248,249,250].

Aging can influence the homeostasis and physiologic functions of the vitreous body [251].

b.
**Aging changes of the vitreous body**


Like all tissues in the human body, during the lifetime cycle, the vitreous body also undergoes age-related changes. The aging of the vitreous is a complex process, which includes biochemical, rheological, and structural changes [252]. Age-related degeneration through structural destabilization of vitreous matrix transforms the vitreous body from a homogeneous and transparent structure to a non-homogenous body with optically attenuated areas [253]. It is well known that a progressive increase in liquified spaces (*synchisis senilis*) as well as optically dense areas with the aggregation of collagen fibers (*syneresis*) occurs with aging of the vitreous body [254]. The liquefaction of the vitreous has been described as early as age 4, increasing to 12.5% at age of 18, to 25% in individuals aged 40–49 years, reaching 62% in individuals aged 80–89 years [254,255]. Both the liquefaction and aggregation are physico-chemical degenerative changes characterized by conformational state dissociation of HA from collagen [256]. The exact mechanisms which lead to these degenerative changes have not been fully elucidated. Many studies suggest two potential mechanisms that cause alterations in the biochemical structure of the vitreous. The first is oxidative-stress induced alteration through reactive oxygen species which contributes to an increased content of free radicals [16]. These radicals then induce HA depolymerization and loss of its MW [16]. A significant decrease in the vitreous HA levels with aging has been reported by Itakura et al. [257]. The second mechanism which takes place by increased enzymatic activity has been reported by Vaughan-Thomas et al. [258]. The significant increase in plasmin(ogen) could potentially activate matrix metalloproteinase which may be involved in the collagen cleavage [258]. Bishop et al. reported an exponential loss of collagen IX during aging [259]. Since collagen type IX acts as a protective shield for collagen type II, its reductions could predispose the vitreous collagen fibrils to fuse and aggregate [259]. The breakdown of fibrillar collagen as well as HA depolymerization causes collapse of the collagen framework and formation of collagen-free, liquid-filled spaces [260]. In these processes, there seem to be plenty of antioxidants (riboflavin, ascorbic acid, glutathione, taurine, crystallin, cysteine, L-tyrosine, human serum albumin, pigment epithelium-derived factor), trace elements including zinc and selenium, and enzymes (superoxide dismutase, glutathione peroxidase and catalase) as protectors of the vitreous matrix stability [16]. In addition to aging, these changes are influenced by other factors such as environmental factors, exposure to sunlight, oxidative stress and HA-collagen interaction [255]. Axial myopia is also an important factor inducing earlier vitreous syneresis which increases with increase in the axial length of the eye [261]. An important factor that accelerates early changes of the vitreous structure seem to be cataract surgery as well. Alterations in the vitreous proteome of pseudophakic eyes, more specifically particles which are likely primarily composed of HA, have been reported by Neal et al. [262]. These changes may be even more dramatic in cases of lesions of the vitreolenticular barrier, which may occur during cataract surgery [263]. In addition to changes in the aging of the vitreous bod, alterations occur also in the vitreoretinal interface. With age-related accumulation of oxidative stress and neurodegeneration, thickening of the internal limiting membrane occurs, which probably causes weakening of the vitreoretinal adhesion [264]. The concurrent vitreous body liquefaction with vitreoretinal adhesion weakening lead to posterior vitreous detachment (PVD). Changes in the vitreous gel structure with collagen fibers’ aggregation and condensation result in patients seeing visually disturbing floaters (*myodesopsia*) [255,264]. A sudden onset of floaters to an even greater extent occurs after acute PVD [264]. The latter can occur in an innocuous way, making the patients to have the perception of floaters and flashes of light, but this event can also be anomalous, resulting in various pathologic manifestations including vitreous hemorrhage, retinal tear, RD, macular hole, epiretinal membrane, as well as vitreo-macular or vitreo-papillary traction syndrome, and it should always be aggressively checked [264].

c.
**The Vitreous Substitutes in Current Use: Advantages and Disadvantages**


Having so many complex functions, it is evident that the vitreous substitutes in use as of today have many disadvantages compared to the healthy vitreous. Table 3 summarizes the advantages and disadvantages of the vitreous substitutes currently in use. Since SiOs constitute the major group of vitreous substitute utilized in case of complex retinal detachments, and since they need a second operation to be removed from the vitreous cavity, they are still a source of concern [86].

The purpose of using SiO as a replacement for the vitreous is to temporarily or permanently tamponade the retina. The relationship between buoyancy, interfacial surface tension, and viscosity determines how the SiO behaves dynamically [86].

The difference in specific gravity between a specific SiO and the aqueous humor is what causes buoyancy. As previously discussed, a vitreous substitute’s ability to float or sink in aqueous humor depends on its specific gravity. Aqueous humor and vitreous humor have somewhat greater specific gravities than water (1.00), but SiO has a slightly lower specific gravity (0.97). As a result, SiO floats inside the vitreous cavity, and buoyancy is the term for the upward force. This force is highest at the apex and gradually decreases to zero at the horizontal meniscus. Consequently, tamponade force arises from the difference in the density between aqueous humor, vitreous humor, and SiO bubble. However, the buoyancy does not act upon a single point, but is spread over a limited area and, for this reason, it produces pressure (force/unit area). Since surface tension represents the forces that tend to keep a bubble intact, it is responsible for the form of liquid droplets. Typically, at 25 °C, it is 40 mN/m for 1000 cSt SiO, which is about one-third of the force produced on an air bubble. Once a SiO bubble has been injected into the eye, several things might affect its surface tension. First, viscosity: the surface tension increases with increasing viscosity. The fact that SiOs with greater viscosities are thought to emulsify less frequently than SiOs with lower viscosities is one of the reasons behind this. Viscoelastic solutions, blood, proteins, lipids, and ionized solutions (e.g., biological fluids) are examples of substances that, if present in the vitreous cavity when a SiO is injected, can lower the surface tension, and thus result in emulsification [80,111,272,273]. Emulsification of SiO is a clinically relevant consequence of intraocular silicone usage. Contaminants, protein surfactants, and low MW components and low viscosity, as well as additional surgical procedures play a role in the occurrence. However, the single most critical factor in the emulsification of SiO is the duration of tamponade and the most common related complications are glaucoma and keratopathy [61].

## 7. Vitreous Substitutes as Drug Delivery Agents

The blood-retina and blood-water barriers segregate the vitreous, complicating medication administration by topical or systemic injection [245]. Therefore, a topically administered drug can penetrate either the cornea or the conjunctiva/sclera, but is rapidly washed out from the anterior chamber and rarely reaches the vitreous chamber [274]. Similarly, a systemically administered drug should reach a high blood concentration to gain access into the vitreous body [275]. On the contrary, intravitreal injection of drugs such as anti-VEGFs can achieve a high concentration in the vitreous body that, however, are relatively short-lived [276]. By utilizing an appropriate vitreous replacement that releases the medication over time, the requirement for repeated intravitreal injections might be decreased or eliminated, which would increase patient comfort and adherence, and other active principles could be employed for use in other chronic eye conditions [277]. The use of a hydrogel-based vitreous substitute can be especially advantageous for both the quick and continuous release of drugs, unlike liquids (BSS and SiO) and gases [180,278]. Such an effect can be achieved through a variety of mechanism that are under investigation and summarized in Figure 6.

## 8. The Ideal Vitreous Substitute

The ideal vitreous substitute should be structurally and functionally identical to the natural vitreous. To keep the ocular structures in the right place and maintain a normal intraocular pressure (IOP), it should possess similar viscoelastic qualities. It should allow for the movement of ions and electrolytes while being optically clear. It should be simple to control and self-renewing to only need one implantation. Additionally, it should not harm other ocular structures, be biocompatible, not degrade over time, be easily accessible for a fair price, and be easy to store [59,60,111].

## 9. Conclusions and Future Perspectives

The endotamponades (aqueous solutions, gases, and silicone oils) utilized on a regular basis for clinical features have demonstrated to be effective in promoting retinal reattachment. However, such endotamponades substantially differ from the characteristics of the human vitreous, resulting in a wide range of complications and therefore failure to achieve the requirement of ideal vitreous substitute. Nevertheless, recent advanced studies have convincingly demonstrated the potential of polymeric hydrogel-based vitreous substitutes, which can allow for a fast and wide-range of therapeutics. Despite this, there are still some limitations that require assessment of the hydrogel-based tamponades before this therapeutic technology is used in clinical settings.

The current vitreous substitutes are based on hydrophobic substances that can only tamponade in one direction, like gases (upward) or HSiO (downward), and cannot completely fill the hydrophilic vitreous cavity where growth factors might aggregate and stimulate proinflammatory processes [279,280]. The hydrogel-based substitutes are hydrophilic substances that can be perfectly in contact with all the parts of the vitreous cavity, and this supports a paradigm shift from hydrophobic to hydrophilic vitreous substitutes [178]. Furthermore, hydrophilic hydrogels have shown an important role as medication repositories and drug delivery devices [180]. Nevertheless, these materials have only been examined under preclinical studies with animal models, and the scarcity of clinical studies is not allowing for the translation of these novel approaches into the clinic.

The vitreous substitutes are excellent surgical devices that allow surgeons to address several vitreoretinal conditions both intra- and post-operatively. The prognosis of many illnesses has improved thanks to contemporary vitrectomy methods and the ability to utilize the large variety of tamponades. The wide selection of vitreous substitutes available based on the vitreoretinal pathology enables surgeons to select the best intraocular tamponade for the underlying condition. In addition, the surgeon can select between several endotamponade features based on the circumstances and length of permanence of the tamponade in the eye. However, it is important to remember that some procedures, as complete removal of all tractions, are essential to the operation’s success.

The choice to use SiO can be made both before and during the procedure performed. The surgeon’s perspective is that the time needed for the tamponade is the first and most crucial aspect to consider. It is essential to evaluate the kind of vitreoretinal illness and the post-operative position prior to the procedure because, if we know that the patient will not be able to maintain a particular post-operative position, it may be preferable to use SiO over a different tamponade. However, even if a permanent tamponade is not planned, it is still feasible to decide during the procedure that SiO is required.

Since vitreoretinal diseases requiring PPV as a treatment of choice are often associated with other ocular conditions such as glaucoma or age-related macular degeneration [281,282,283], PPV could become a surgical opportunity for treating the concurrent chronic conditions with an artificial vitreous substitute capable of storing and releasing a controlled amount of pharmacologically active molecules.

For a very long time, it was believed that the vitreous played only a small part in the structure and function of the eye. Its significance in preserving an ideal environment for the retina and the other surrounding tissues was not realized until the development of new surgical procedures. Due to this, initial research efforts were directed at developing a vitreous replacement that had comparable physical and biological characteristics to the genuine vitreous, with little success. The necessity for a long-lasting, high quality vitreous substitute will hopefully be met by new classes of vitreous tamponades, but still, future research should aim at achieving a product capable of blending the quality of SiO and gas tamponades without their risks and side effects.

## 10. Methods of Literature Search

A search of the MEDLINE database was conducted using the key words vitreous substitutes, silicone oil, tamponade, vitreoretinal surgery, and retinal detachment. Additional references were obtained from the bibliographies of these references. References spanned the period of 1911 to 2022 and were evaluated for their pertinence to the topic, with special consideration given to papers that detailed a correlation between biomaterial properties and clinical correlation of the various substitutes in a translational perspective. Articles that were repetitious or decidedly nonclinical were omitted from consideration.

## Figures and Tables

**Figure 1 ijms-24-03342-f001:**
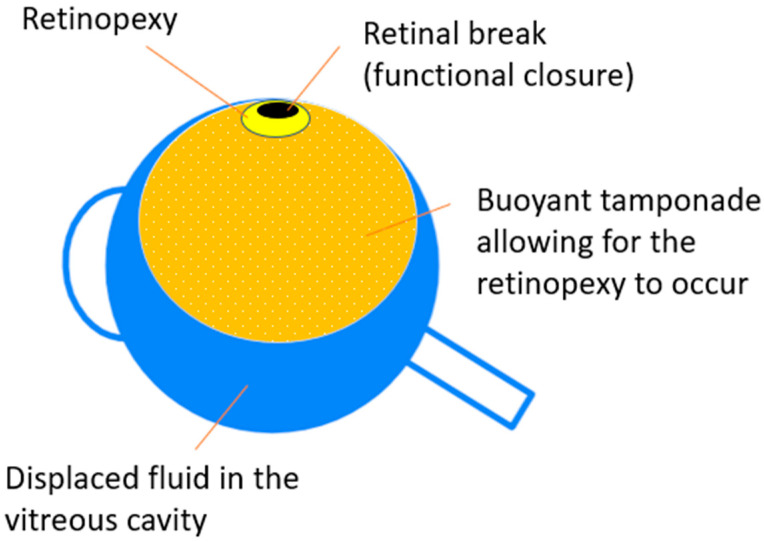
General representation of the mechanism of buoyant vitreous tamponades. The tamponade allows for the chorioretinal contact and adherence to occur, while displacing the intravitreal fluid, potentially containing proinflammatory cytokines. The bubble tamponades the retinal break and blocks the intravitreal fluid from passing through the break into the subretinal space until retinopexy takes place.

**Figure 2 ijms-24-03342-f002:**
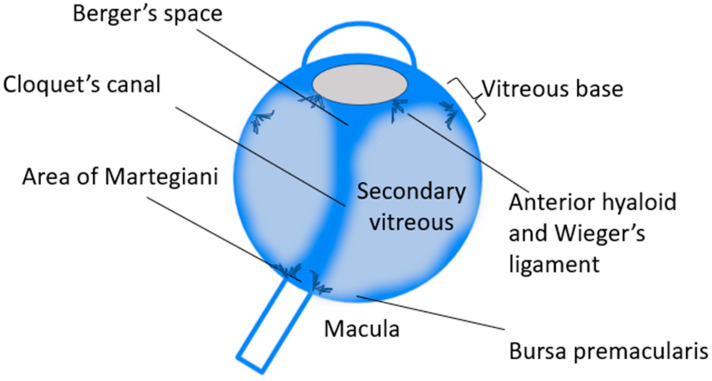
Anatomy of the vitreous body and adjacent structures. In order to adapt to the surrounding anatomical structures, similarly to water in a complexly shaped container, the vitreous body must bridge and connect the tissues, leaving spaces and adhesion points whose function is often incompletely known and difficult to reproduce by vitreous substitutes. The embryological vitreous also testifies to the final anatomy, and the Cloquet’s canal represents the major remnant due to the regression of the hyaloid artery [17].

**Figure 3 ijms-24-03342-f003:**
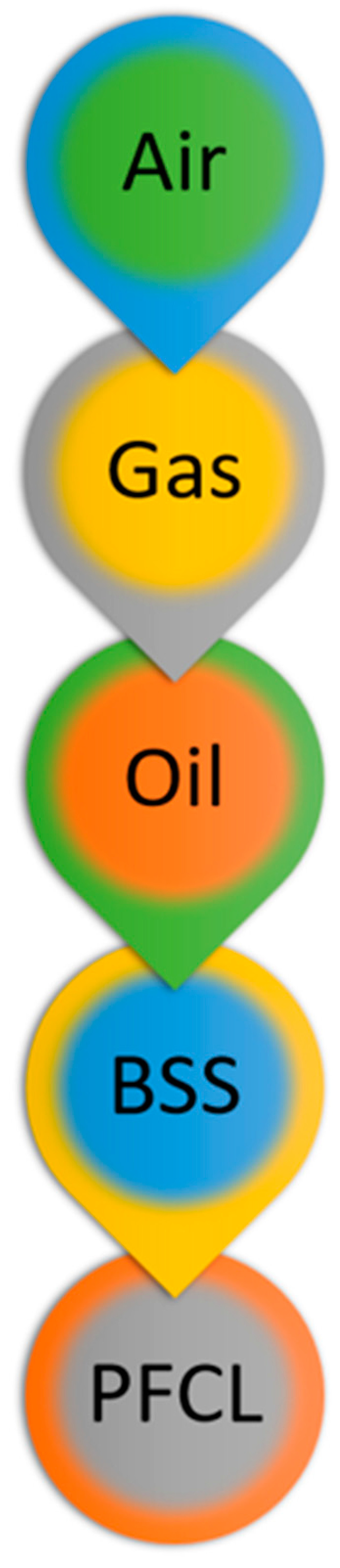
The vitreous substitutes are hereby represented as they would arrange in their resting position with respect to each other in a hypothetical vitreous chamber, according to their buoyancy property. Abbreviations: BSS, balanced salt solution; PFCL, perfluorocarbon liquid.

**Figure 4 ijms-24-03342-f004:**
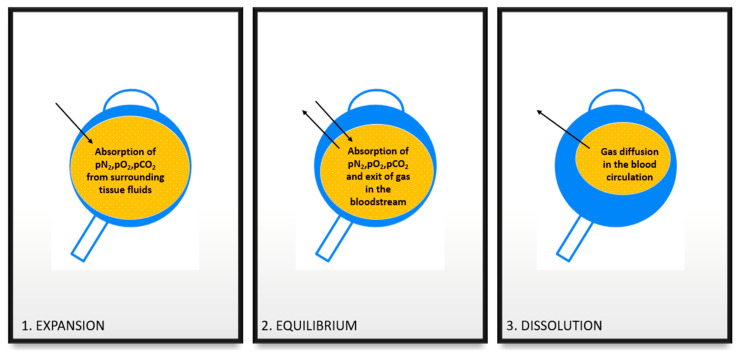
Schematic of the three phases of a gas bubble injected into the vitreous cavity. (**1**) Expansion, occurring 6–8 h postoperatively with a net acquisition of pN_2_, pO_2_, and pCO_2_ into the gas tamponade bubble from the surrounding tissue fluids. (**2**) Equilibrium, occurring in the following days according to the gas type, where the absorption of the above-mentioned molecules is compensated by the slippage of gas into the blood circulation. (**3**) Dissolution corresponds to the permanence time of gas into the eye when the volume of the gas bubble shrinks until it is completely absorbed into the blood circulation.

**Figure 5 ijms-24-03342-f005:**
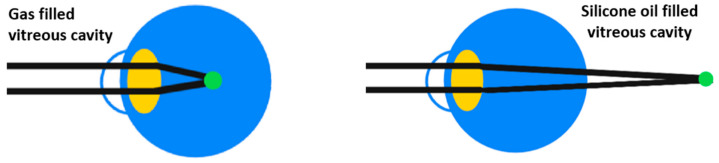
A gas filled vitreous chamber (**left**) has a higher difference in refractive index between the posterior face of the crystalline lens than vitreous and an even higher difference than silicone oil (**right**). The result is a surgically induced myopia in case of a gas filled vitreous cavity and a surgically induced hyperopia in case of a silicone oil filled vitreous cavity.

**Figure 6 ijms-24-03342-f006:**
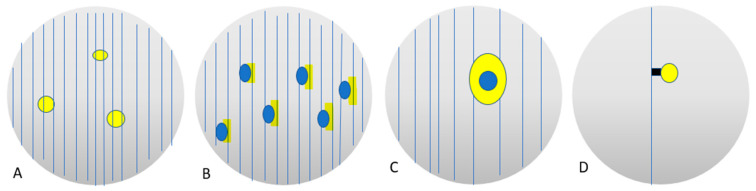
Mechanisms of drug release in the vitreous. (**A**) Slow drug release through polymer’s mechanical entrapment. (**B**) Slow drug release through polymer’s electrostatic interaction. (**C**) Slow drug release through polymer’s mechanical entrapment via micro- or nano-particle. (**D**) Slow drug release through polymer’s chemical binding.

**Table 1 ijms-24-03342-t001:** Properties of the gases commonly utilized in vitreoretinal surgery.

Type of Gas	Molecular Weight (g/mol)	Maximal Expansion (h)	Duration in the Vitreous Chamber	Non-Expansile Concentration	Capacity to Expand (Times)
Air	29	No expansion	5–7 days	No expansion	0
SF_6_	146	24–48	1–2 weeks	20%	2
C_2_F_6_	138	36–60	4–5 weeks	16%	3
C_3_F_8_	188	72–96	6–8 weeks	12%	4

Abbreviations: SF_6_ = sulfur hexafluoride; C_2_F_6_ = perfluoroethane; C_3_F_8_ = perfluoropropane.

**Table 2 ijms-24-03342-t002:** (**a**) Overview of hydrogels-based vitreous substitutes. (**b**) Overview of hydrogels-based tamponades-based vitreous substitutes.

(a)					
Hydrogels	Polymer Content [%]	Refractive Index	Light Transmittance [%]	In Vivo Studies	Ref.
Gellan and hyaluronic acid	1	-	85–95	Not conducted	[192]
Methacrylated gellan gum	1	-	-	Not conducted	[224]
Hyaluronic acid	1	1.34	-	rabbits	[184]
	3	1.34	-	rabbits	[185]
	1–2.2	1.32–1.34	-	rabbits	[182]
	1	1.32–1.33	-	rabbits	[225]
	1	1.34	75–91	Not conducted	[226]
Peptide gel	0.10	1.33	96.7	rabbits	[227]
**Synthetic polymers**					
Polyvinyl alcohol methacrylate	9	-	-	Not conducted	[194]
Polyvinyl alcohol	7	-	-	macaques	[195]
	4	-	85	Not conducted	[196]
	5	-	-	Not conducted	[197]
	1–7	1.34	93	rabbits	[198]
	4	1.34	-	Not conducted	[199]
Poly (ethylene glycol)	5	1.34	-	rabbits	[228]
Acrylic acid and acrylamide	1.25–1.75	-	-	Not conducted	[201]
Poly N-acryloy| glycinamide-polycarboxybetaine acrylamide	1.60	1.34	93.2	rabbits	[203]
**Smart hydrogels**					
WTG-127	-	-	89.3	rabbits	[229]
Poly (ethylene glycol)	25	1.35	>90	Not conducted	[230]
	10	1.33	-	rabbits	[231]
	0.4–0.7	-	-	rabbits	[232]
Sulfobetaine methacrylamide and acryloyl cystamine monomers	5	-	>90	rabbits	[233]
Gellan and poly (methacrylamide-co-methacrylate)	0.65–1.29	1.34–1.34	87.6–94	rabbits	[191]
(**b**)					
**Hydrogels**	**Polymer Content [%]**	**Refractive Index**	**Light Transmittance [%]**	**In Vivo Studies**	**Ref.**
**Smart hydrogels**					
Methacrylic acid, methylacrylamide, and bismethacryloyloystamine	1–1.4			Not conducted	[234]
Polymethacrylamide and poly-methacrylate	0.9–1.8	1.33–1.33	>95	Not conducted	[235]
Hydroxypropyl chitosan and alginate dialdehyde	1–3	1.33	>80	rabbits	[189]
poly(ε-caprolactone)-based (PCL) thermogel	3–12	1.34–1.34		rabbits	[207]
Polyhydroxyalkanoate (PHA) thermogel	2–20		<90	rabbits	[236]
Poly (ethylene glycol) methacrylate and poly (ethylene glycol) diacrylate	0.75–5.7	1.34–1.34	>90	Not conducted	[210]
Gellan and poly(methacrylamide-co-methacrylate-co- bis (methacryloy))lorstamine)		1.34–1.34	>83	rabbits	[237]

**Table 3 ijms-24-03342-t003:** Advantages and disadvantages of the vitreous substitutes available in the current clinical practice.

Vitreous Substitute	Advantages	Disadvantages	Ref.
BSS	Transparent, physiologic refractive index, non-toxic, such as aqueous humor and vitreous	Low interfacial tension, density nearly equal to 1	[7,265]
Air	Chemically inert, non-toxic, inexpensive, colorless, low density, high interfacial tension, no need for a second operation for removal	Very short duration inside the vitreous cavity, low refractive index (1.0008)	[7,114]
Gas (SF_6_, C_2_F_6_, C_3_F_8_)	Colorless, odorless, non-toxic, high interfacial tension and buoyancy	Short duration inside the vitreous cavity, induction of cataract development, high IOP, need for positioning	[7,266,267]
PFCL	Transparent, high density, moderate interfacial tension	Only approved for intra-operative use due to high reactivity	[268,269]
SFAs	Inert, colorless, physiologic refractive index	Low specific gravity; cataract; emulsification. Epiretinal membrane development	[99,164,270]
Silicone Oil	Moderate interfacial tension, Transparent, inert	Cataract, glaucoma, corneal toxicity, silicone retinopathy risk; second surgery for its removal	[86,271]

## Data Availability

Data are available on reasonable request by the corresponding authors.

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
