# Peer review of "Vitreous Substitutes from Bench to the Operating Room in a Translational Approach: Review and Future Endeavors in Vitreoretinal Surgery"

_ijms, 2023, doi:10.3390/ijms24043342_

Round 1

Reviewer 1 Report

Dear authors, 

Despite the fact that there are several publications on the topic, this review provides a comprehensive and updated overview. The article is well-written in English and includes appealing illustrations and tables. However, your review is well conducted and highlighted some interesting features, but still needs significant improvement. My specific comments are: 

- There are a lot of bulleted lists, e.g. "Cellular and biomolecular architecture of the vitreous body" should be added to the section "The Anatomy of the Vitreous Body:..";

- I would also remove from the title the words " from Bench to the Operating Room”.

- You should pay attention to the references (e.g. n 26, 262), please rearrange the paper according to the “reference list and citations guide”.

- 267 and 268 are duplicated references, please rearrange them.

- reference numbers placed in square brackets [ ] must be placed before the punctuation please rearrange the paper.

Kind regards

Author Response

Q - There are a lot of bulleted lists, e.g., "Cellular and biomolecular architecture of the vitreous body" should be added to the section "The Anatomy of the Vitreous Body".

A – The bulleted lists have now been reduced, and the two sections have been merged. Thank you for the suggestion for improvement.

Q - I would also remove from the title the words " from Bench to the Operating Room”.

A – We agree with the reviewer - the title has been modified accordingly.

Q - You should pay attention to the references (e.g., n 26, 262), please rearrange the paper according to the “reference list and citations guide”.

A – Thank you for pointing this out and sorry for the confusion - the references have now been modified accordingly.

Q - 267 and 268 are duplicated references, please rearrange them.

A – Again, thank you for pointing this out, the references have been modified accordingly.

Q - reference numbers placed in square brackets [ ] must be placed before the punctuation please rearrange the paper.

A – Thank you, the paper has been rearranged accordingly.

Reviewer 2 Report

Line 69. Might add “where the vitreous body is most adherent to the underlying retina and ciliary epithelium” after vitreous base

Line 182. Add “s” to direction.

Line 186. Delete “is” before floats.

Line 188. Add “RPE and” after onto the

Line 192. “cytokines”

Line 196 “immersed”

Figure 2. Need to define BS and PFCL in the drawing.

Line 259, rather than harmless, might say “non-toxic” since overfilling can lead to severe pressure rises

Line 275. “expand” Change cases to “cases”

Line 276. Define IOP if not done elsewhere

Line 297. Change “for” to “by”.

Line 298. Add “it” after makes.

Line 309 “example”

Line 310. “units”

Line 391. The authors write “a. Gases” but do not follow up with b. or c. categories. This needs to be corrected.

Line 395. Change “costs” to “cost”.

Line 418. “retina”

Line 423. Delete “the” before “surgical”.

Line 424. Change to “choroidal effusion following underfilling of the vitreous chamber. This can be overcome by a slightly higher………………………”

Line 432. Delete “the” before “others”.

Line 454. Add “toxicity” before “stems”.

Line 546. Add “due to” before “the toxicity”

Line 557. Delete “-“ after “drug”.

Line 569. Substitute “are” for “showed to be”.

Line 639. Substitute “there seem to be” for “seem to be involved”.

Line 648. Replace “seem to be also” with “seems to be”

Line 652. “aging”

Lines 660-1. Might reword this. Would not want the reader the believe flashes and floaters should not be aggressively checked.

Line 666. Do the authors believe there is no fluid (aqueous) flow through the vitreous gel when they say stagnant?

Line 749. Substitute “have” for “on”.

Lines 759-60. Delete “, which form an interface with the hydrophilic vitreous cavity,”.

Line 765. Delete the sentence.

Line 767. Insert “surgeons” after “allow”

Line 772. Insert “can” for “would also be able to”

Line 773.  “endotamponade”

Line 790. Substitute “small” for “little”.

Nice job on the sciencee nature of the subject.

Author Response

Line 69. Might add “where the vitreous body is most adherent to the underlying retina and ciliary epithelium” after vitreous base.

A: Thank you for the suggestion, we have added the suggested sentence to the text.

Line 182. Add “s” to direction.

A: Thank you for the suggestion, we have added the suggested letter.

Line 186. Delete “is” before floats.

A: Thank you for the suggestion, we have deleted it.

Line 188. Add “RPE and” after onto the

A: Thank you for the suggestion, we have added it.

Line 192. “cytokines”

A: Thank you for the suggestion, we have corrected it.

Line 196 “immersed”

A: Thank you for the suggestion, we have corrected it.

Figure 2. Need to define BS and PFCL in the drawing.

A: Thank you for the suggestion, we have defined them.

Line 259, rather than harmless, might say “non-toxic” since overfilling can lead to severe pressure rises

A: Thank you for the suggestion, we have modified it.

Line 275. “expand” Change cases to “cases”

A: Thank you for the suggestion, we have modified them.

Line 276. Define IOP if not done elsewhere

A: defined in line 114.

Line 297. Change “for” to “by”.

A: Thank you for the suggestion, we have modified it.

Line 298. Add “it” after makes.

A: Thank you for the suggestion, we have added it.

Line 309 “example”

A: Thank you for the suggestion, we have modified it.

Line 310. “units”

A: Thank you for the suggestion, we have modified it.

Line 391. The authors write “a. Gases” but do not follow up with b. or c. categories. This needs to be corrected.

A: Thank you for the suggestion, we have corrected it.

Line 395. Change “costs” to “cost”.

A: Thank you for the suggestion, we have corrected it.

Line 418. “retina”

A: Thank you for the suggestion, we have corrected it.

Line 423. Delete “the” before “surgical”.

A: Thank you for the suggestion, we have corrected it.

Line 424. Change to “choroidal effusion following underfilling of the vitreous chamber. This can be overcome by a slightly higher………………………”

A: Thank you for the suggestion, we have rephrased the sentence.

Line 432. Delete “the” before “others”.

A: Thank you for the suggestion, we have corrected it.

Line 454. Add “toxicity” before “stems”.

A: Thank you for the suggestion, we have added it.

Line 546. Add “due to” before “the toxicity”

A: Thank you for the suggestion, we have added it.

Line 557. Delete “-“ after “drug”.

A: Thank you for the suggestion, we have added it.

Line 569. Substitute “are” for “showed to be”.

A: Thank you for the suggestion, we have added it.

Line 639. Substitute “there seem to be” for “seem to be involved”.

A: Thank you for the suggestion, we have corrected it.

Line 648. Replace “seem to be also” with “seems to be”

A: Thank you for the suggestion, we have corrected it.

Line 652. “aging”

A: Thank you for the suggestion, we have checked and both ‘’ageing’’ and ‘’aging’’ are correct; we used “aging” throughout the text uniformly.

Lines 660-1. Might reword this. Would not want the reader the believe flashes and floaters should not be aggressively checked.

A: Thank you for the suggestion, we have reworded it.

Line 666. Do the authors believe there is no fluid (aqueous) flow through the vitreous gel when they say stagnant?

A: We do not mean that there is no fluid flow through the vitreous gel when we say stagnant but we mean that the structure of the vitreous body is relatively idle compared to aqueous humor which has a high turnover. The stability of the vitrous is also potentiated at the end of that sentence: “its structure is rather stable throughout the lifetime”.

Line 749. Substitute “have” for “on”.

A: Thank you for the suggestion, we have corrected it.

Lines 759-60. Delete “, which form an interface with the hydrophilic vitreous cavity,”.

A: Thank you for the suggestion, we have corrected it.

Line 765. Delete the sentence.

A: Thank you for the suggestion, we have corrected it.

Line 767. Insert “surgeons” after “allow”

A: Thank you for the suggestion, we have corrected it.

Line 772. Insert “can” for “would also be able to”

A: Thank you for the suggestion, we have corrected it.

Line 773.  “endotamponade”

A: Thank you for the suggestion, we have corrected it.

Line 790. Substitute “small” for “little”.

A: Thank you for the suggestion, we have corrected it.

Nice job on the science nature of the subject.

A: Thank you so much for your kind indications and thorough review.

Reviewer 3 Report

This review article proposes to identify areas of need for translational research into vitreous substitutes by comprehensively reviewing the clinical data on existing vitreous substitutes.  A strength is that the authors are thorough in describing the various substances that have been studied.

1. However, this paper is written with a very poor understanding of the clinical application of vitreous substitutes.  The authors propose that there should be a move away from hydrophilic, gel-like substitutes, since they are not useful in retinal detachment repair.  However, most hydrophilic gels have been developed for the express purpose of serving as a repository for intravitreal medications rather than for retinal detachment repair.  This is a significant clinical need that could be highly beneficial that is significantly under appreciated and not described by the authors.  Or do they confuse the two applications of vitreous substitutes?  It is not clear.

2. This review adds very little to the field.  A very comprehensive and well-written review was performed by Teri Kleinberg in 2011, and the only new addition to the field has been more synthetic hydrogels.  If this review went into more depth into recent developments and focused needs for research, rather than focusing so much on the substitutes that have been around for 10-30 years and then simply dismissing all hydrophilic gels out of hand, it might be a better contribution to the literature.

3. The authors state several things that are frankly untrue and incorrect, primarily surrounding the function of a tamponade, and demonstrate a poor understanding of why retinal detachment repair works.  There are too many for me to detail, so I will just list some examples below.  I suggest that the authors either recruit a co-author who has a solid fundamental understanding of vitreoretinal clinical and surgical practice and the mechanisms of successful retinal detachment repair and drug delivery, or else that at the very least, the authors read the relevant sections of the textbook Michel's retinal detachment, ideally the 1998 version, .  https://www.amazon.com/Retinal-Detachment-Ronald-G-Michels/dp/0801634172

and learn about the purpose and clinical need for long-standing intravitreal drug repositories.

A few examples:

(Line 14-15) The two crucial functions of these substitutes are their ability to displace aqueous humor from the retinal surface.

Aqueous humor is not in contact with the retinal surface.  Liquefied vitreous is what needs to be kept away from retinal breaks.

(Line 15-16) and to keep the retina adherent to the RPE.

Vitreous substitutes are not glue, and as such, they do not keep the retina adherent to the RPE.  Things like laser retinopexy, cryopexy, or the RPE fluid pump keep the retina adhered.  In contrast, substitutes like gas block fluid from flowing through the break and detaching the retina and the buoyancy can temporarily push the retina against the RPE.

I quote from Michels retinal detachment book, p 481.  The surface tension effects of an intravitreal gas bubble effectively tamponade retinal breaks and prevent liquid vitreous from passing through the break into the subretinal space... the tamponade effect depends on the bubble preventing intravitreal fluid from coming into contact with the break and also on the bubble itself not passing through the break... Functional closure of retinal breaks by intraocular gas tamponade results in absorption of subretinal fluid and flattening of the entire retina against the pigment epithelium.

(Lines 137-138)  The third objective is to limit the diffusion of proliferative and 137 inflammatory cytokines into the vitreous cavity so to prevent PVR development.

There are actually high levels of inflammatory cytokines in the sub-silicone oil fluid, and PVR occurs regularly with silicone oil in place, arguing against any anti-inflammatory benefits.

https://www.ncbi.nlm.nih.gov/pmc/articles/PMC5454016/

(Lines 234-235) On the other hand, the tendency of a substance to emulsify and disperse into droplets over time is also dependent on its viscosity.

Although this is true in general of substances outside of the human body, in clinical practice, emulsification occurs pretty equally in 1000 cs and 5000 cs SO.

(Lines 276-277). The same volumetric expansion can 276 also be caused by higher-than-normal blood partial pressure of nitrous oxide, oxygen 277 and carbon dioxide and carbon monoxide.  

If oxygen causes gas expansion, why does every patient who has surgery and is given oxygen not have gas expansion?

4. The article is structured strangely in a way that seems to maximize redundancy.  I suggest combining section 4. Chemical Properties of the Currently Available Tamponades and Clinical Correlates and section 5. Classification of the Vitreous Substitutes: General Principles, Past and Future Attempts and Clinical Correlates.  This will reduce the number of times the reader has to hear the basics about conventional silicone oil.

Author Response

This review article proposes to identify areas of need for translational research into vitreous substitutes by comprehensively reviewing the clinical data on existing vitreous substitutes.  A strength is that the authors are thorough in describing the various substances that have been studied.

A: We thank the reviewer for the thorough critique and pointing out the strength of the paper, but also its weaknesses, which we now go through in a point-by-point fashion below.

  1. However, this paper is written with a very poor understanding of the clinical application of vitreous substitutes.  The authors propose that there should be a move away from hydrophilic, gel-like substitutes, since they are not useful in retinal detachment repair.  However, most hydrophilic gels have been developed for the express purpose of serving as a repository for intravitreal medications rather than for retinal detachment repair. This is a significant clinical need that could be highly beneficial that is significantly underappreciated and not described by the authors.  Or do they confuse the two applications of vitreous substitutes?  It is not clear.

A: Thank you for raising these points out. The paper is actually co-authored by 1 vitreoretinal surgeon with over 40 years experience (R.B.), 2 with over 30 years (I.S-J., X.L.) and 1 with over 15 years of experience (G.P.).

We are not suggesting by any means that there should be a move from hydrophilic, gel-like substitutes – this has now been better explained in the text, please see the discussion that says: ‘’The hydrogel-based substitutes are hydrophilic substances that can be perfectly in contact with all the parts of the vitreous cavity and this supports that it is required a paradigm shift from hydrophobic to hydrophilic vitreous substitutes’’. In fact, we agree with the Reviewer about the use of such substitutes, in particular as repositories for intravitreal medications. We hope with the new clarification, such substitutes will be shown in a non-confusing or non-ambiguous way to the Reviewer and the future readers.

  1. This review adds very little to the field.  A very comprehensive and well-written review was performed by Teri Kleinberg in 2011, and the only new addition to the field has been more synthetic hydrogels.  If this review went into more depth into recent developments and focused needs for research, rather than focusing so much on the substitutes that have been around for 10-30 years and then simply dismissing all hydrophilic gels out of hand, it might be a better contribution to the literature.

A: By no means we wanted to minimize the previously written reviews and research in the field. Indeed, tamponades are a continuously updated and highly developing field, beyond the review of Kleinberg et al (2011), which we certainly recognize and know well.

It was, however, not our intention to miss onto the hydrophilic gels/substitutes, which is now better explained in the Discussion section.   

  1. The authors state several things that are frankly untrue and incorrect, primarily surrounding the function of a tamponade, and demonstrate a poor understanding of why retinal detachment repair works.  There are too many for me to detail, so I will just list some examples below.  I suggest that the authors either recruit a co-author who has a solid fundamental understanding of vitreoretinal clinical and surgical practice and the mechanisms of successful retinal detachment repair and drug delivery, or else that at the very least, the authors read the relevant sections of the textbook Michel's retinal detachment, ideally the 1998 version, .  https://www.amazon.com/Retinal-Detachment-Ronald-G-Michels/dp/0801634172

and learn about the purpose and clinical need for long-standing intravitreal drug repositories.

A: We thank the Reviewer for pointing out these fundamental issues. The co-authors with their extensive experience in vitreoretinal surgery and teaching of residents nationally and internationally, feel that the confusing parts or obstacles can be resolved within the present authorship composition. With true respect to the Michel’s textbook (also familiar to the more and less experienced authors), we are fully respecting the postulates established in it. Our response to the few examples follows below. 

A few examples:

(Line 14-15) The two crucial functions of these substitutes are their ability to displace aqueous humor from the retinal surface.

A: The reviewer is right about this unclarity, which we have now explained better in the text. In brief, we did not state that anterior segment aqueous moves to the back of the eye, or the tamponades in any way displace such fluid from the detached retina surface.

Aqueous humor is not in contact with the retinal surface.  Liquefied vitreous is what needs to be kept away from retinal breaks.

A: Indeed, we fully agree with this statement, and we have modified the explanation in the text accordingly.

(Line 15-16) and to keep the retina adherent to the RPE.

Vitreous substitutes are not glue, and as such, they do not keep the retina adherent to the RPE.  Things like laser retinopexy, cryopexy, or the RPE fluid pump keep the retina adhered.  In contrast, substitutes like gas block fluid from flowing through the break and detaching the retina and the buoyancy can temporarily push the retina against the RPE.

A: Again, we agree with the reviewer that vitreous substitutes are not glue, and that the later stated procedures (laser retinopexy, cryopexy) or the RPE fluid pump are the mechanisms keeping the retina adherent. To better describe the mechanism, the reviewer is referring to, i.e. the buoyancy forces, so we therefore made a new Figure/schematic showing how those forces work. We hope our response will satisfy the reviewer’s request, now.

I quote from Michels retinal detachment book, p 481.  The surface tension effects of an intravitreal gas bubble effectively tamponade retinal breaks and prevent liquid vitreous from passing through the break into the subretinal space... the tamponade effect depends on the bubble preventing intravitreal fluid from coming into contact with the break and also on the bubble itself not passing through the break... Functional closure of retinal breaks by intraocular gas tamponade results in absorption of subretinal fluid and flattening of the entire retina against the pigment epithelium.

A: We hope our figure on the surface tension effects of tamponades will suffice the criticism of the reviewer and contribute to better clarity of the description of the mechanisms.

(Lines 137-138)  The third objective is to limit the diffusion of proliferative and 137 inflammatory cytokines into the vitreous cavity so to prevent PVR development.

There are actually high levels of inflammatory cytokines in the sub-silicone oil fluid, and PVR occurs regularly with silicone oil in place, arguing against any anti-inflammatory benefits.

https://www.ncbi.nlm.nih.gov/pmc/articles/PMC5454016/

A: We fully agree with the Reviewer that such objectives should be met. In fact, we are saying exactly that, and we have now added a figure on the role of proliferative and inflammatory cytokines present in the remaining fluid phase not covered by the tamponades.

(Lines 234-235) On the other hand, the tendency of a substance to emulsify and disperse into droplets over time is also dependent on its viscosity.

Although this is true in general of substances outside of the human body, in clinical practice, emulsification occurs pretty equally in 1000 cs and 5000 cs SO.

A: We fully agree with the reviewer and to clarify the point we have added a sentence that stresses this concept.

(Lines 276-277). The same volumetric expansion can 276 also be caused by higher-than-normal blood partial pressure of nitrous oxide, oxygen 277 and carbon dioxide and carbon monoxide.  

If oxygen causes gas expansion, why does every patient who has surgery and is given oxygen not have gas expansion?

A: This is a good question, thank you. The answer is that we have to consider the molecular oxygen partial pressure in the blood stream. It is clear that this effect by molecular oxygen can become clinically significant only in cases associated with absorption of other molecules, and when the partial pressure of the molecular oxygen is considerably higher than physiologic. To stress the point, we have now elaborated in a sentence, adding the following references:

  1. The article is structured strangely in a way that seems to maximize redundancy.  I suggest combining section 4. Chemical Properties of the Currently Available Tamponades and Clinical Correlates and section 5. Classification of the Vitreous Substitutes: General Principles, Past and Future Attempts and Clinical Correlates. This will reduce the number of times the reader has to hear the basics about conventional silicone oil.

A: Thank you so much for the suggestion. The two sections have been created to stress different characteristics of belonging to the same compounds. However, we agree with the Reviewer that the redundancy must be minimized and, to do so, we have paid attention to avoid that in the revised version of the manuscript.

Round 2

Reviewer 3 Report

The authors did not substantially address my comment #1, the need to clarify that some vitreous substitutes function as tamponades in retinal detachment repair and others function as drug delivery agents.  They made some minor changes but overall the entire paper is unclear because these two functions are not clearly distinguished.

I still maintain that this review is not adding much to the field (as in previous comment #2) because it is primarily a description of things that have been around for more than 10 years and were well described in a previous review.

Regarding my previous comment #3, the authors have satisfactorily corrected most of the inaccuracies regarding the function of tamponades.

Regarding my previous comment #4, the authors did not restructure the review to improve readability and reduce redundancy.

Author Response

Q: The authors did not substantially address my comment #1, the need to clarify that some vitreous substitutes function as tamponades in retinal detachment repair and others function as drug delivery agents.  They made some minor changes but overall, the entire paper is unclear because these two functions are not clearly distinguished.

A: Thank you. To answer this question, we have now added a paragraph dedicated to tamponades as drug delivery agents with a new figure (Figure 6).

Q: I still maintain that this review is not adding much to the field (as in previous comment #2) because it is primarily a description of things that have been around for more than 10 years and were well described in a previous review. Regarding my previous comment #3, the authors have satisfactorily corrected most of the inaccuracies regarding the function of tamponades.

A: Thank you. We hope we could be as inclusive as possible and provide overview of the old and newer literature reports in our review. With that, we then hope to have achieved even a small contribution to the field, with some future perspectives also mentioned.

Q: Regarding my previous comment #4, the authors did not restructure the review to improve readability and reduce redundancy.

A: We have made a new attempt to revise the English in the manuscript. The structure or division into physical and chemical properties in the paper, we believe, should remain. In fact, even though some overlapping is apparent, it was meant to help the reader connect or verse between the topics, and understand why the molecules that make up the tamponade agent behave in such a way; it also allows to keep the paragraphs in a self-standing way.
